**Funding:** The project is supported by FISM - Fondazione Italiana Sclerosi Multipla – cod. 2016/B/3 and financed or co-financed with the '5 per

# A group resilience training program for people with multiple sclerosis: Study protocol of a multi-centre cluster-randomized controlled trial (multi-READY for MS)

**Ambra Mara Giovannetti** [1,2]*, **Kenneth Ian Pakenham**[2], **Giovambattista Presti**[3], **Maria Esmeralda Quartuccio**[4], **Paolo Confalonieri**[5], **Roberto Bergamaschi**[6], **Monica Grobberio**[7], **Massimiliano Di Filippo**[8], **Mary Micheli**[9], **Giampaolo Brichetto**[10,11], **Francesco Patti**[12], **Massimiliano Copetti**[13], **Paola Kruger**[14], **Alessandra Solari**[1]

1 Unit of Neuroepidemiology, Fondazione IRCCS Istituto Neurologico Carlo Besta, Milano, Italy, 2 School of Psychology, Faculty of Health and Behavioural Sciences, University of Queensland, Brisbane, QLD, Australia, 3 Kore University Behavioral Lab, Faculty of Human and Social Sciences, Università degli Studi di Enna 'Kore', Enna, Italy, 4 Department of Neurosciences, San Camillo- Forlanini Hospital, Roma, Italy, 5 MS Centre, Unit of Neuroimmunology and Neuromuscular Diseases, Fondazione IRCCS Istituto Neurologico Carlo Besta, Milano, Italy, 6 IRCCS Fondazione Mondino, Pavia, Italy, 7 Laboratorio di neuropsicologia, UOSD psicologia clinica e UOC neurologia, ASST Lariana, Como, Italy, 8 Centro Malattie Demielinizzanti e Laboratori di Neurologia Sperimentale, Clinica Neurologica, Università degli Studi di Perugia, Perugia, Italy, 9 Dipartimento Riabilitazione ASLUmbria2, Foligno, Italy, 10 AISM Rehabilitation Service of Genoa, Italian Multiple Sclerosis Society, Genova, Italy, 11 Scientific Research Area, Italian MS Society Foundation, Genova, Italy, 12 Neurology Clinic, Multiple Sclerosis Centre, University Hospital Policlinico Vittorio Emanuele, Catania, Italy, 13 Unit of Biostatistics, Fondazione IRCSS Casa Sollievo della Sofferenza, San Giovanni Rotondo, Italy, 14 Patient Expert, EUPATI Fellow (European Patients Academy for Therapeutic Innovation) Italy, Roma, Italy

* ambra.giovannetti@istituto-besta.it

## Abstract

### Introduction

REsilience and Activities for every DaY (READY) is an Acceptance and Commitment Therapy-based group resilience-training program that has preliminary empirical support in promoting quality of life and other psychosocial outcomes in people with multiple sclerosis (PwMS). Consistent with the Medical Research Council framework for developing and evaluating complex interventions, we conducted a pilot randomized controlled trial (RCT), followed by a phase III RCT. The present paper describes the phase III RCT protocol.

### Methods and analysis

This is a multi-centre cluster RCT comparing READY with a group relaxation program (1:1 ratio) in 240 PwMS from eight centres in Italy (trial registration: isrctn.org Identifier: ISRCTN67194859). Both interventions are composed of 7 weekly sessions plus a booster session five weeks later. Resilience (primary outcome), mood, health-related quality of life, well-being and psychological flexibility will be assessed at baseline, after the booster session, and at three and six month follow-ups. If face-to-face group meetings are interrupted because of COVID-19 related-issues, participants will be invited to complete their

mille' public funding. The funders had no role in study design, data collection and analysis, decision to publish, or preparation of the manuscript.

**Competing interests:** AMG reports a grant from Rehabilitation in Multiple Sclersis (RIMS) during the conduct of the study. AS reports grants from Fondazione Italiana Sclerosi Multipla (FISM), during the conduct of the study; personal fees from Biogen Idec, Merck Serono, Novartis, Almirall, and Excemed. As a staff member of the University of Queensland and co-Author of the READY program, KP receives royalties from UniQuest for commercial (not research) licensing arrangements entered into by third parties who want to deliver the program. PC has received honoraria for speaking or consultation fees from Novartis and Biogen, has received funding for travel to attend scientific events or speaker honoraria from Merck Serono, Biogen Idec, Teva and Roche. He has also received institutional research support from Merk-Serono, Novartis and Roche. He is also principal investigator in clinical trials for Biogen, Merck Serono, and Roche. This does not alter our adherence to PLOS ONE policies on sharing data. All the other authors reports no competing interests.

intervention via teleconferencing. Relevant COVID-19 information will be collected and the COVID-19 Peritraumatic Distress scale will be administered (ancillary study) at baseline and 3-month follow-up. Analysis will be by intention-to-treat to show superiority of READY over relaxation. Longitudinal changes will be compared between the two arms using repeated-measures, hierarchical generalized linear mixed models.

## Conclusion

It is expected that his study will contribute to the body of evidence on the efficacy and effectiveness of READY by comparing it with an active group intervention in frontline MS rehabilitation and clinical settings. Results will be disseminated in peer-reviewed journals and at other relevant conferences.

## Introduction

Multiple sclerosis (MS) is a chronic disease that affects the central nervous system through processes of demyelination and degeneration that ultimately cause neuronal damage and axonal loss [1]. The prevalence of MS is rising globally [2]. The most recent study on the global prevalence of MS estimated that 2.8 million people worldwide have MS [3]. Italy, with more than 125,500 persons with MS (PwMS), has one of the highest prevalence rates [4]. Because MS onset is usually between 20 and 40 years old [5], the disease has the potential to severely impact most life domains [6, 7]. Moreover, PwMS often have to cope with uncertainty about disease progression, loss of function, changes in life roles and a variety of symptoms [8]. These stressors can evoke a deep sense of personal vulnerability [9] and thwart the illness adjustment process [10]. Indeed, evidence shows that PwMS have poorer quality of life (QoL) than healthy controls and people with other chronic diseases [11, 12]; with a recent meta-analysis reporting a lifetime prevalence of 30.5% for depression and 22.1% for anxiety symptoms in this population [13]. In addition, research evidence supports the association between psychological stress and subsequent relapses in MS, with the occurrence of stressful life events purported to lead to a greater risk of relapse [14].

Evidence suggests that resilience plays a key role in alleviating the adverse effects of stress and sustaining mental health in adversity [15]. It entails the process of negotiating, managing and adapting to significant stressors or trauma through drawing on internal (i.e. mindfulness, acceptance, cognitive flexibility and active coping), and external (i.e. social support, financial capital and community services) resources [16]. When facing adversity and stressful situations, people with lower resilience have a higher risk of experiencing poorer QoL, and greater distress and relational difficulties [17], and adopting unhealthy behaviors, which in turn are likely to negatively affect physical health [18]. As reported by Strike and Steptoe, poor psychosocial functioning and exposure to prolonged stress are likely to adversely impact physical health through physiological stress reactions such as hypertension, blood pressure increases, pro-inflammatory cytokines and the development of metabolic syndrome [19]. Given the evidence showing PwMS have lower resilience than the general population and people with other chronic diseases [20], they are particularly vulnerable to the adverse effects of stress. Therefore, evidence-based interventions aimed at fostering resilience are pivotal in helping them positively cope with their illness-related stressors and promote well-being and QoL.

Resilience-training interventions have been shown to promote a range of positive psychosocial outcomes in people with chronic illnesses [21], including QoL, anxiety, depression, perceived stress and well-being in adults with cancer [22–24], congenital heart disease [25], diabetes [26], neurofibromatosis [27], and MS [28]. However, the quality of many of these studies is sub-optimal [29], and Chmitorz et al. identified three main methodological problems: definitions of resilience as trait or a composite of resilience factors rather than as a process; the use of psychometrically weak instruments; the omission of one or more key study design elements (e.g., a priori sample size calculation, adequate comparator, sufficient baseline diagnostics, long-term follow-up assessment, adverse effects assessments, participant satisfaction evaluation, and multi-centre study designs) [30].

In the last decade, an Australian team has developed and tested an Acceptance and Commitment Therapy (ACT)-based group resilience-training program called REsilience and Activities for every DaY (READY). The READY program was initially applied in a workplace setting [31, 32], and then adapted and successfully implemented with different health conditions: cancer [33, 34], diabetes [35], and MS [28]. The Australian case series study on READY for MS showed that the program had beneficial impacts on resilience, QoL, depression, stress and protective factors (managing difficult thoughts, values and acceptance) in PwMS [28].

READY is informed by ACT, which is a third wave cognitive behavior therapy that aims to promote psychological flexibility. According to ACT, psychological flexibility involves behaving consistently with personal values even in the presence of psychological discomfort [36]. Psychological flexibility is established through the following six processes: (1) acceptance–openness to experience, (2) cognitive defusion–observing thoughts rather than taking them literally, (3) present moment awareness–mindfulness, (4) self-as-context–contact with a sense of self that is continuous and provides flexible perspective taking, (5) values–freely chosen personally meaningful life directions, (6) committed action–values-guided effective action [36]. The READY program uses these six core ACT processes to target five empirically supported resilience protective factors: cognitive flexibility (defusion), acceptance, meaning, social connectedness and values-based action.

ACT is as an empirically supported intervention for the promotion of mental health across a wide range of contexts [37–40], and in people with various health conditions [33–36, 41–47]. A recent review of meta-analyses demonstrated that ACT is superior to inactive controls (e.g. waitlist, placebo), treatment as usual, and most active intervention conditions (with the exception of other cognitive behavioral therapy-based intervention) [48].

Based on the empirical support for ACT and the READY program, we developed a research protocol to evaluate the efficacy of READY for MS (here-on called READY) in Italy. We designed the project following the Medical Research Council (MRC) framework for developing and evaluating complex interventions, which entails a multi-phased approach involving a pre-clinical research phase and a final phase in which the intervention is introduced into the health service [49]. Hence, the project consisted of two phases: 1) a pilot randomized controlled trial (RCT) with a nested qualitative study [50]; 2) a multi-centre phase III RCT, which is the focus of the present study. The second phase had an ancillary study that evaluated the impact on psychologists of their training in READY [51] and the effectiveness of READY delivered via frontline Italian health services by psychologists trained in the intervention [52].

Results from the pilot study showed that although the READY program was well accepted by PwMS and suitable for Italian clinical settings, there were no between-arm differences in any patient-reported outcomes. Three methodological limitations may explain the absence of statistical superiority of READY over relaxation: small sample size, short follow-up, and a ceiling effect with the primary outcome (QoL). Qualitative data showed that participants viewed READY as superior to relaxation; a finding that converged with four non-significant statistical

trends (resilience, psychological flexibility, acceptance and defusion) supporting READY efficacy. Consistent with the ACT psychological flexibility framework, participants viewed their improvements in resilience and health-related QoL as being due to the acquisition of skills related to the six core ACT processes [50].

The trial steering committee (TSC) and an international expert panel discussed the pilot study findings in two dedicated meetings (January 2020). The panel discussion was structured on the "PICO" (Population, Intervention, Comparator, Outcomes) format [53]. Panel recommendations are reported in Box 1.

> ## Box 1. Panel recommendations
>
> - The READY and control interventions should be delivered by different facilitators.
>
> - Control intervention facilitators should not have had prior training in mindfulness or ACT nor receive such training for the duration of the study.
>
> - A six-month post-intervention follow-up should be scheduled in addition to the 3-month follow-up (primary endpoint).
>
> - Following Chmitorz et al.'s suggestion that resilience should be the primary outcome measure in resilience interventions, and based on the trend differences observed in the pilot RCT, the Connor-Davidson Resilience Scale 25 (CD-RISC 25) was chosen as the primary outcome.

## Materials and methods

This is a multi-centre cluster RCT assessing the superiority of READY over relaxation, with three and six-month follow-ups. The study follows the CONSORT guidelines for RCTs on social and psychological interventions (CONSORT-SPI 2018) [54]. The study protocol was designed following the Standard Protocol Items: Recommendations for Interventional Trials (SPIRIT) guidelines (S1 Appendix) [55], and The SPIRIT-PRO Extension [56]. The SPIRIT schedule of enrolment is reported in Fig 1, and the CONSORT flowchart in Fig 2.

We hypothesize that compared to the control arm (relaxation), READY participants would show greater improvements on the primary outcome of resilience and on the secondary outcomes of mood, health-related QoL, well-being and psychological flexibility. The primary endpoint is the between-arm difference from baseline (T0) to three-month follow-up (T2) changes in resilience scores.

The protocol received ethical clearance from the ethics committee of the Fondazione IRCCS Istituto Neurologico Carlo Besta (15th April 2020, internal ref: 71; amendment approved 18th November 2020, internal ref: 78) and it has been evaluated by each participating centre's ethics committee. After obtaining Ethics Committee approvals from all the participating centres, the study will be run at the MS/rehabilitation units of eight university hospitals, research hospitals, general hospitals or community health services across Italy.

### Eligibility criteria

Participant inclusion criteria are: MS diagnosis [57]; age $\geq$18 years; written informed consent; CD-RISC 25 score $<$ 83, which indicates that the person could still improve his/her level of

| | STUDY PERIOD | | | | | |
|---|---|---|---|---|---|---|
| | Enrolment | Allocation | Post-allocation | | | |
| TIMEPOINT | $-t_1$ | 0 | Baseline (T0) | Post-intervention (T1) | 3-month follow-up (T2) | 6-month follow-up (T3) |
| **ENROLMENT:** | | | | | | |
| Eligibility screen | X | | | | | |
| Informed consent | X | | | | | |
| Allocation | | X | | | | |
| **INTERVENTIONS:** | | | | | | |
| READY for MS | | | ●━━━━━━━━━● | | | |
| Relaxation | | | ●━━━━━━━━━● | | | |
| **ASSESSMENTS:** | | | | | | |
| Demographic data clinical record | | | X | | | |
| Resilience (CD-RISC 25) | X* | | | X | X | X |
| Anxiety and depression (HADS) | | | X | X | X | X |
| Affect (PANAS) | | | X | X | X | X |
| MS-related QoL (MSQOL-54) | | | X | X | X | X |
| Health-related QoL (The EQ-5D-3L) | | | X | X | X | X |
| Well-being (MHC-SF) | | | X | X | X | X |
| Psychological Flexibility (MPFI) | | | X | X | X | X |
| Satisfaction with the intervention | | | | X | X | |
| COVID-19 Peritraumatic Distress | | | X | | X | |

**Fig 1. Schedule of enrolment, interventions and assessments.** Note. * CD-RISC 25 is administered after informed consent is signed.

resilience; able to attend group sessions, and fluent Italian speaker. PwMS will be excluded from the study if one or more of the following criteria are met: severe cognitive compromise (Mini Mental State Examination <19); psychosis or other serious psychiatric conditions; psychotherapy in the preceding six months; prior formal training in mindfulness methods or current meditation practice; severe suicidality, including ideation, plan and intent; one or more relapses in the previous month; corticosteroid treatment during the previous month; other serious medical conditions in addition to MS; current pregnancy; MS diagnosis for less than three months.

## Recruitment and trial procedures

A flyer, which includes a general overview of the study and contact details, will be sent via e-mail to PwMS by the MS Centre team. People who show interest in participating in the study will be contacted by the study Principal Investigator (PI) or centre PI. Subsequently, one trained clinical psychologist (READY facilitator) will make an appointment with those patients who met the inclusion criteria and agreed to participate in the study, and check all eligibility criteria. She/he will send an e-mail to the participant with a link to the website containing the set of questionnaires. The assessment will last about 55 minutes at each time-point (additional 20 minutes at T1 and T2 for completing satisfaction questionnaires). Each centre will collect information on the number of people approached, screened, and eligible prior to random assignment, including reasons for non-enrolment. Participants will then be assigned to the READY or relaxation in a 1:1 ratio using the method of minimization (two factors: Centre and CDRISC

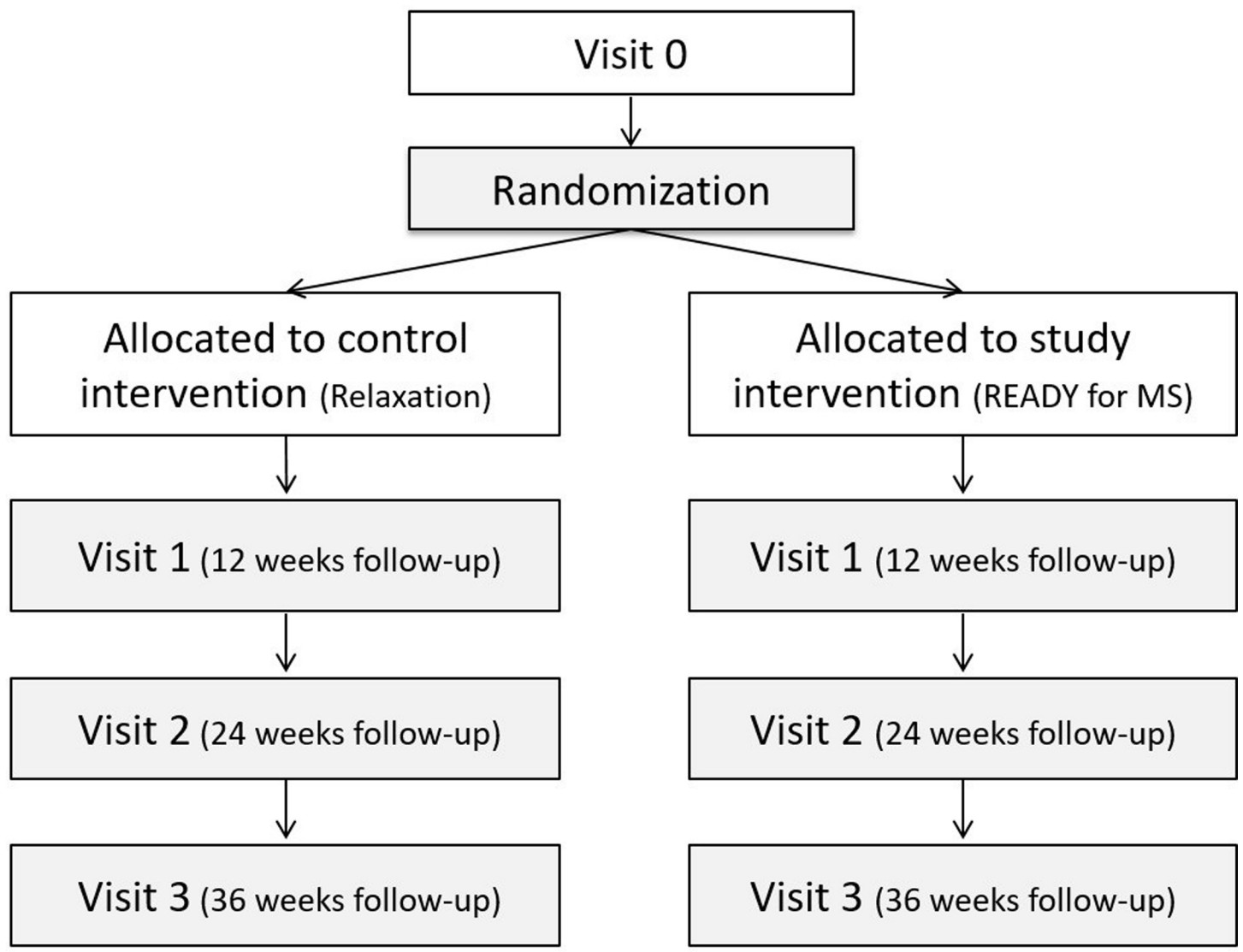

**Fig 2. Study flowchart.**

score $< 50$ and $\geq 50$) [58]. Treatment assignment will be provided by an independent randomization unit, using a computer-based algorithm [59]. Confirmation e-mails will be sent to the study PI. The interventions will start within two weeks of the baseline assessment.

## Confidentiality

All study-related information will be stored securely at the study site. Data collection, process, and administrative forms will be identified by a coded ID number only to maintain participant confidentiality. All records that contain names or other personal identifiers, will be stored separately from study records identified by code number. All local databases will be secured with password-protected access systems. Forms, lists and any other listings that link participant ID numbers to other identifying information will be stored in a separate, locked file in an area with limited access. Audio recordings of the sessions will be deleted immediately after intervention fidelity assessments are completed (no later than one week after a session).

### Pre-study interview and informed consent (baseline visit, T0)

During the pre-study evaluation each potential participant will receive comprehensive verbal and written information about the nature and purpose of the study. Written, signed informed consent will be obtained in accordance with the Declaration of Helsinki and the GCP Guidelines of the EU.

### Assessments

Patient reported outcome measures (PROMs) will be administered at T0, after the booster session (T1, 12 weeks after baseline visit), at three (T2), and six month follow-ups (T3). Participants will complete the purpose-built Participant Satisfaction Questionnaire at T1 and T2. They will receive an email with the link to the website containing the set of questionnaires. Additional process data will capture participant attendance and homework completion, and facilitator perspectives on a weekly basis. At T0 the patient's referring neurologist will report the following clinical information on the case report form: age at MS diagnosis; expanded disability status sale (EDSS) score [60]; MS course (relapsing remitting, primary progressive, secondary progressive); and ongoing disease modifying treatment. Neurologists will update the occurrence of new relapses at each time-point.

### Interventions

Each intervention group will have 8–10 participants. A total of 24 groups will be run (12 READY and 12 relaxation). The READY facilitators are member of "The psychologist network of the Italian Multiple Sclerosis Association (AISM)" who successfully completed "The ACT and be READY for MS Training Program" [51]. The relaxation facilitators are psychologists not involved in "The ACT and be READY for MS Training Program" and with no prior training in ACT or mindfulness interventions. Each group will be run in a dedicated room of the corresponding centre. Participants and facilitators will be asked not to disseminate their intervention information or materials. Each centre will organize READY and relaxation sessions at different times/days in order to prevent possible contact between READY and relaxation participants.

 **READY.** READY is an adult ACT informed group resilience-training program, in which resilience is metaphorically described as a shield composed of five life domains (thinking, feeling, doing, relations and being). Within each life domain an empirically supported key resilience protective factor is highlighted and targeted by the intervention. These protective factors reflect one or more of the core ACT processes. Although for ease of presentation and understanding, the shield contains seemingly separate facets, the domains and protective factors are dynamic and overlapping. The ACT processes impact multiple domains and protective factors. In particular, mindfulness and self-as-context occupy a central pivoting role due to their diffuse and synergistic psychosocial effects (Fig 3) [61]. Content of the seven weekly sessions is as follows: an introductory module (Introduction to the READY Resilience Model), five modules focusing on the six ACT processes (Mindfulness, Acceptance, Cognitive Defusion, Self-as-context, Values and Meaningful Action), and a review module (Review and Future Planning). The booster session provides a review of the program content (S2 Appendix). The program has a facilitator manual, participant workbook, and audio recordings of mindfulness exercises. Throughout the program, participants are encouraged to share their progress and experience of applying the READY strategies and techniques. It incorporates a blend of psychoeducation and experiential exercises, combined with readings and homework exercises that participants practice between sessions [28, 50].

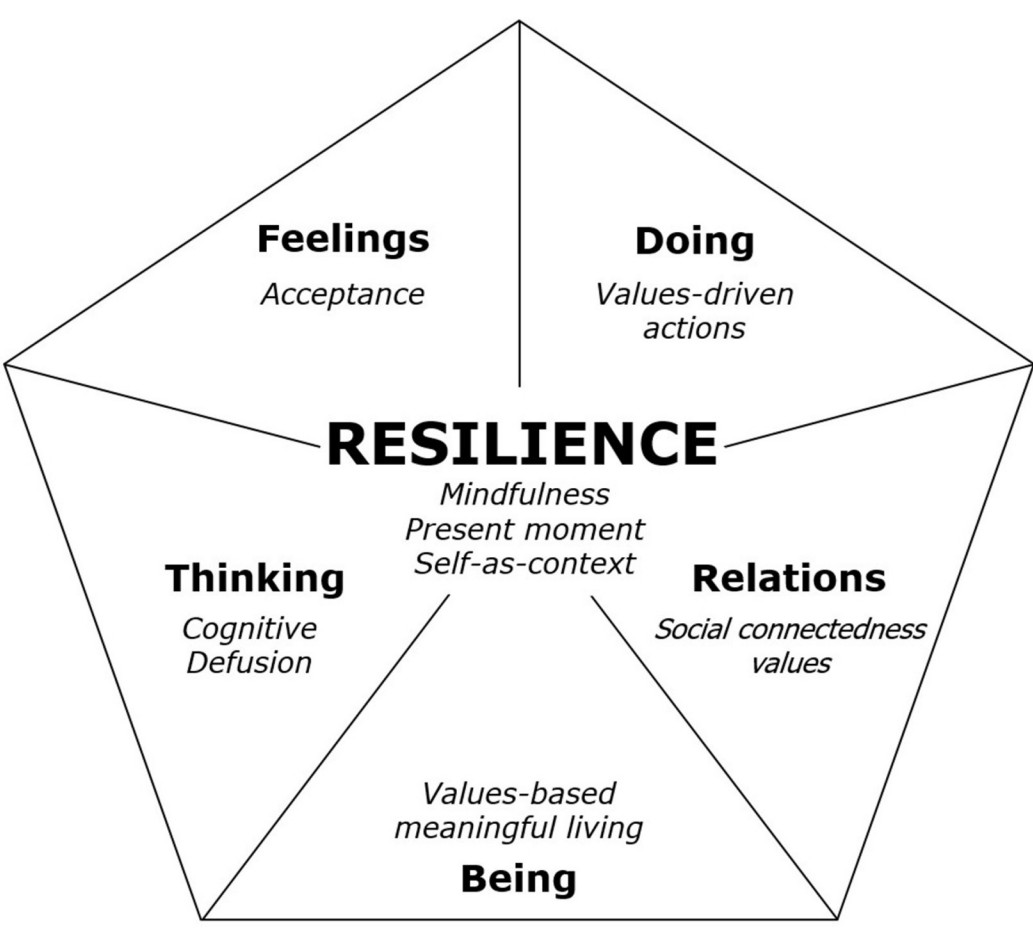

**Fig 3. The READY resilience shield.** Note. Life domains are in bold, ACT processes reflecting resilience protective factors (italics) are reported in the relevant life domain (shield clove) or in the centre of the shield if they have a central pivoting role.

**Relaxation.** The control intervention consists of a group relaxation program based on autogenic training [62]. The program matches the study intervention in number of sessions (seven) and schedule (S3 Appendix) but not in session content and length (1-hour). The program has a facilitator manual, participant workbook, and audio recordings of relaxation exercises.

**Intervention fidelity and supervision.** Each session (READY and relaxation) will be audio-recorded and the file sent via e-mail to the study PI immediately after recording. The procedure for monitoring intervention fidelity includes the following actions.

After each session, facilitators complete a purpose-build Session Fidelity Checklist and The Acceptance and Commitment Therapy Fidelity Measure (ACT-FM; only READY facilitators) [63], and send them to the study PI. The Session Fidelity Checklist lists the sequence of components for each session as reported in the Facilitator Manual. It also includes a section for clinical notes. Two versions are available, one for READY (S4 Appendix), and one for the relaxation program (S5 Appendix).

The 25-item ACT-FM evaluates adherence to the ACT therapeutic style. Items are rated on a 4-point Likert scale. It yields two scores that reflect the overarching dimensions of ACT consistent and inconsistent therapist behaviour. Each of these dimensions has four subscales: 1)

ACT Consistent Therapist Stance; 2) ACT Inconsistent Therapist Stance; 3) ACT Consistent Open Response Style; 4) ACT Inconsistent Open Response Style; 5) ACT Consistent Aware Response Style; 6) ACT Inconsistent Aware Response Style; 7) ACT Consistent Engaged Response Style; 8) ACT Inconsistent Engaged Response Style [63].

After each session the study PI will review these materials to check for self-reported discrepancies with the manual and/or inconsistencies with ACT principles (only READY facilitators). In the case of self-reported discrepancy (or inconsistency with ACT), the study PI will contact the facilitator and discuss the session. The study PI will be available for supervision anytime.

Two audio-recordings randomly selected for each facilitator (one from session 2 or 3 and one from sessions 4 to 7) will be assessed by the study PI using the Session Fidelity Checklist and ACT-FM (only for READY) [63]. READY recordings will also be independently assessed by another ACT expert (TSC member), and in the case of a discrepancy in assessors' ratings, they will discuss the relevant data until a consensus is reached. If low intervention fidelity is identified, the study PI will contact the facilitator to critically discuss the session and the quality of her/his facilitation and the facilitator's next session will also be assessed.

Low fidelity will be determined if at least one of the following criteria are satisfied: A discrepancy with the content manual in more than one section of the Session Fidelity Checklist; For READY facilitators an ACT-FM score $< 5$ for the Stance Consistent section (items 1–4) and a score $< 11$ for the other sections combined (these criteria were defined after discussion with the ACT-FM Authors).

## Patient and public involvement statement

PwMS and an AISM member were involved at several stages of developing the multi-phased project. We received input on the READY and relaxation interventions from participants in the pilot study via both questionnaires and personal interviews [50]. The results of the pilot study were discussed in a dedicated meeting involving the TSC and an expert panel and were used to design the present RCT. The AISM Director of Health Care Professional and Client Services unit had been a member of the TSC until June 2020. An expert MS patient is now a member of the TSC and a co-author of the present paper. We will disseminate key study findings to PwMS with assistance from AISM.

## Outcome measures

**Primary outcome measure.** The Connor-Davidson Resilience Scale 25 (CD-RISC 25) is used to assess psychological resilience. It is composed of 25 items, each rated on a 5-point scale (0–4), with higher scores reflecting greater resilience. The scale has demonstrated good psychometric properties (i.e. its internal consistency, test–retest reliability, and convergent and divergent validity) [64]. The CD-RISC scores have been shown to increase with treatments hypothesized to enhance resilience [65]. It also obtained the highest rating in a methodological review of resilience measures [16].

**Secondary outcome measures.** The following PROMs will be administered in the order they are presented.

*Anxiety and depression.* The Hospital Anxiety and Depression Scale (HADS) is a well-validated measure that consists of two seven-item subscales to assess anxiety and depressive symptoms. Higher scores indicate higher levels of depressive and anxiety symptoms [66]. Unlike many similar measures, the HADS excludes somatic symptoms of anxiety and depression, which may overlap with physical illness symptoms [66].

*Affect.* The Positive and Negative Affect Schedule (PANAS) consists of two 10-item mood scales and it is a self-report measure of positive and negative affect. Respondents rate the extent

to which they have experienced each particular emotion during the last two weeks on a 5-point Likert scale. The PANAS has been shown to be a reliable and valid measure of affect [67].

*MS-related QoL.* The Multiple Sclerosis Quality of Life-54 (MSQOL-54) is a MS-specific measure of Health-related QoL. It comprises the generic 36-item Short-Form (SF-36), plus 18 MS-specific items [68]. The 54 items are organized into 12 multi-item and two single item subscales. As for the SF-36, two composite scores (Physical and Mental Health Composite) are derived by combining scores of the relevant subscales. The MSQOL-54 has well documented content, construct and discriminative validity and reliability [68, 69].

*Health-related QoL.* The EQ-5D-3L is a preference-based health-related QoL measure with one question for each of the five dimensions: mobility, self-care, usual activities, pain/discomfort, and anxiety/depression. It also includes a Visual Analog Scale for perceived health status ranging from 0 (the worst possible health status) to 100 (the best possible health status) [70].

The 14-item Mental Health Continuum Short Form (MHC-SF) measures social, emotional and psychological well-being. Respondents rate the frequency of various experiences in the past month on a 6-point Likert scale. The MHC-SF has shown good psychometric properties [71].

*Psychological flexibility.* The Multidimensional Psychological Flexibility Inventory (MPFI) is a 60-item self-reported questionnaire assessing psychological flexibility and inflexibility. Previous studies confirmed the two-factor second order model in both the original English MPFI and the Italian MPFI version [72, 73]. To limit participants' burden in questionnaires completion, only the psychological flexibility subscale (30 item) will be included in the present study. The psychological flexibility subscale includes 6 dimensions of the Hexaflex model (i.e. acceptance, mindfulness, self-as-context defusion, contact with values, and committed action) [72]. Each dimension is evaluated via 5 items. Items are rated on a 6-point Likert scale, from 1 "never true" to 6 "always true". Higher scores indicated higher level of psychological flexibility. The questionnaire in general and its subscales have demonstrated good psychometric properties in both clinical and nonclinical samples [40, 74–76]. Moreover, findings showed that MPFI subscales have good validity in terms of responsiveness to change over time [72].

*Satisfaction with the intervention.* Four purpose-built questionnaires (two for each intervention) explore satisfaction with READY or relaxation at post-intervention and 3-month follow-up.

Adherence to the intervention. After each session facilitators will collect information on participant's attendance and ask the participants to rate their level of commitment to homework activities.

We will use the Italian versions of the HADS [77], PANAS [78], MSQOL-54 [79], EQ-5D-3L [80], MHC-SF [81], and MPFI [73]. For the CD-RISC 25 we will use the unpublished Italian version [courtesy of Davidson].

## Data analysis

**Sample size calculation.**   The sample size calculation was based on the pilot study results [50], which showed a mean change in the CD-RISC 25 at 3-month follow-up in the READY arm equal to 15.61 (SD 13.63, n = 18) vs. a mean change in the relaxation arm equal to 5.95 (SD 14.11, n = 19). The intra-cluster correlation coefficient (ICC) was equal to 0.08.

A sample size of 12 clusters per study arm (total number of clusters = 24) with 10 PwMS per cluster (total sample size 240) achieves 94.6% power to detect a mean difference of 9.66 between the two arms [82]. We made the following assumptions: SD of change in the CD-RISC 25 at 3-month of 14.53; ICC of 0.10; and alpha value of 0.05.

Based on these figures (i.e., total sample size 240, 24 clusters, 10 subjects per cluster), we computed the study power considering different drop-out scenarios (S6 Appendix).

**Statistics.** Analyses will be carried out in accordance with the pre-specified statistical analysis plan and performed by study personnel blind to the participants' assignment and to those providing the interventions.

Baseline variables' standardized mean differences (for clustered data) between arms will be computed to measure potential unbalance.

Longitudinal changes will be analyzed using repeated measures hierarchical (patients nested in clusters) generalized linear mixed models, accounting for the cluster effect (using random intercepts for clusters) and adjusted for those baseline covariates resulting unbalanced between arms. As reported above, the between-arm difference from baseline (T0) to three-month follow-up (T2) changes in resilience scores is the primary endpoint. To mitigate the risk of inflating Type I error given the small number of clusters, we will use the Kenward-Roger degrees of freedom correction, which does not rely on the assumption of fixed cluster sizes [83].

All group comparisons will be carried out according to the intention-to-treat principle. That is, participants will be analysed in the arm (READY or relaxation) to which they were assigned. We will use multiple imputations for missing data where appropriate (i.e. drop-out, missing item/questionnaire) [84]. In addition, we will carry out a per-protocol analysis and assess the sensitivity of the results to excluding patients who missed three or more READY sessions.

The study will be overseen by an independent data and safety monitoring committee (DSMC) consisting of three members with expertise in biostatistics, complex interventions, and psychology. No interim analyses will be conducted. The DSMC will monitor accrual and retention of participants. Protocol amendments are made in consultation with the DSMC.

## Strategies for limiting COVID-19 impacts on the study

Conducting a study that involves face-to-face group interventions during the COVID-19 pandemic exposes it to the risk of having to abort interventions due to the emergence of local COVID-19 infection 'hot-spots'. The TSC carefully considered three options. First, pausing the study until the pandemic situation is under control. Due to the uncertainties of predictions about when the pandemic will be more manageable, this option was excluded. Second, delivering the interventions online. This option was also excluded because it would violate the MRC framework guidelines by nullifying the pilot RCT which, as purposed, provided data that informed the design of the present multi-centre RCT. Third, retaining the group delivery of both interventions with an option for teleconference delivery in the case of COVID-19 infection threats. This option was accepted by the TSC and involved a set of actions, summarized in Box 2.

---

### Box 2. TSC action plan for limiting COVID-19 impacts on the study

- The study PI will have weekly contact with the participating centres in order to monitor changes in the local spread of COVID-19 and to take prompt infection control actions as necessary in accordance with guidelines from the Italian Government and local health authorities.

- In the case of one or more participating centres not being able to start the study up to four months after the commencement of enrolling participants in other centres, the active centres will run the remaining groups in their place (competitive enrolment).

---

- If face-to-face group meetings are interrupted because of COVID-19 related-issues (e.g., local lockdown), all affected participants will be invited to complete their intervention via teleconferencing and the assessments as scheduled.

- Relevant COVID-19 information will be collected (i.e., COVID-19 infected; family member infected; COVID-19 impacts on employment; risk perceptions of COVID-19 infection because of MS; perceived additional burden of MS treatment/rehabilitation because of COVID-19).

- During data analysis a sensitivity analysis will be undertaken to explore any effects of the teleconference intervention delivery relative to the face-to-face delivery.

The study results will be published in peer-reviewed journals, presented at conferences and a lay summary will be sent to participants. The TSC will suggest topics for presentation or publication and will circulate them to the PI of each participating centre. Topics suggested by a member of the participating centre should be approved by the TSC and the person making the suggestion may be considered as the lead author.

The study was registered on the ISRCTN registry (isrctn.org Identifier: ISRCTN67194859) the 14th May 2020.

## Discussion

The primary purpose of this trial is to assess the effects of a brief, structured group resilience-training intervention on resilience in PwMS over the nine-month study period. A secondary purpose is to assess the effects of the intervention on participant's mood, QoL, well-being and psychological flexibility. Considering that the intervention was developed in Australia and only recently applied in Italy with preliminary empirical support [50], it is important to further investigate the efficacy of READY in the Italian context. Recent literature reviews on resilience have emphasized the need for more methodologically rigorous research on the effects of resilience-training interventions [29, 30]. The design of the present study not only overcomes all the limitations reported in these reviews, it adheres to the MRC framework for developing and evaluating complex interventions [49], and it also aligns with all the recommendations for future research identified by Öst in his systematic review and meta-analysis on the efficacy of ACT interventions with the exception of a follow-up of at least one year [85].

Within this methodological context, this study will contribute to the body of evidence on the efficacy and effectiveness of READY by comparing it with an active group intervention in frontline MS rehabilitation and clinical settings. It is expected that READY will cultivate targeted resilience protective factors that will help PwMS effectively manage MS-related stressors. Moreover, the READY program is relatively brief and highly structured, two characteristics that increase its affordability and ease of dissemination.

It should also be noted that the psychological flexibility processes that underpin the READY intervention have been shown to protect people from the adverse mental health impacts of COVID-19 [86, 87]. For this reason, the Italian version of the COVID-19 Peritraumatic Distress scale will be administered (and completed on a voluntary basis) at baseline and 3-month follow-up (primary endpoint) as part of an ancillary study [86, 88].

### Limitations and measures to minimize bias

Two study limitations are noted. Blinding of patients is not possible due to the type of the study interventions. The READY and relaxation interventions are similar in frequency and

number of sessions, but not in duration. Despite these limitations the following measures will be used to increase methodological rigor. The statistical analyses will be performed by study personnel blind to the participants' randomization to the two intervention conditions and to those providing the interventions. Electronic versions of the study PROMs will be used to ensure the data entered is of high quality. The TSC will monitor adherence to the study protocol and overall study quality. Finally, an independent DSMC will oversee the study procedures, recruitment, and data flow. The study PI and another ACT expert will monitor the intervention fidelity audio recordings and ratings of READY and relaxation sessions. Facilitators will only receive detailed information on the group intervention (READY or relaxation) they conduct. For this reason, facilitator meetings will be run separately for READY and relaxation facilitators. Members of the Clinical Psychology Expert Panel will also be available to discuss any issues facilitators have in relation to delivering the interventions.

## Supporting information

**S1 Appendix. SPIRIT checklist.**
(DOCX)

**S2 Appendix. READY sessions.**
(DOCX)

**S3 Appendix. Relaxation sessions.**
(DOCX)

**S4 Appendix. READY for MS session fidelity checklist.**
(DOCX)

**S5 Appendix. Relaxation session fidelity checklist.**
(DOCX)

**S6 Appendix. Drop-out scenarios.**
(DOCX)

**S1 File. Clinical study protocol.**
(PDF)

## Acknowledgments

Author want to thank Prof. Marta Bassi, Dr. Andrea Giordano, Prof. Stefan Gold, Prof. Christopher Heesen and Dr. Jana Pöttgen for their precious feedback on the study protocol.

 **Collaborators**

 Multi_READY for MS Trial Steering Committee: AMG, AS, KIP, GP, PK. Independent Data and Safety Monitoring Committee: S Gold, M Bassi, MP Sormani. Data Management and Analysis Committee: M Copetti, AG, AMG, AS. Clinical Psychology Expert Panel: AMG, KIP, GP and J Pöttgen. Centres and investigators: Fondazione IRCCS Istituto Neurologico Carlo Besta, Unit of Neuroepidemiology: AG, AMG, AS. Fondazione IRCCS Istituto Neurologico Carlo Besta, Unit of Neuroimmunology and Neuromuscular Diseases, Multiple Sclerosis Centre: PC, Rui Quintas, Milda Černiauskaitė. San Camillo-Forlanini Hospital, Roma: Carla Tortorella, MEQ. AISM Rehabilitation Service of Genoa, Italian Multiple Sclerosis Society, Genova: GB, Miranda Giuntoli, Annalisa Garaventa. Neurology Clinic, Multiple Sclerosis Centre, University Hospital Policlinico Vittorio Emanuele, Catania, Italy: FP, Eleonora Chisari, Chiara Vona. Laboratorio di neuropsicologia, UOSD psicologia clinica e UOC neurologia, ASST Lariana: MG, Samuela Turati. Centro Sclerosi Multipla, Divisione di Neurologia

Generale, IRCCS Fondazione Istituto Neurologico Nazionale C. Mondino di Pavia: RB, Ambrogia Ornella Riolo, Marta Picascia. Dipartimento Riabilitazione ASLUmbria2: MM, Serena De Bigontina. Centro Malattie Demielinizzanti e Laboratori di Neurologia Sperimentale, Clinica Neurologica, Università di Perugia: MDF, Giuliana Costantini, Luciana Ciaccassassi.

## Author Contributions

**Conceptualization:** Ambra Mara Giovannetti, Kenneth Ian Pakenham, Alessandra Solari.

**Data curation:** Ambra Mara Giovannetti.

**Formal analysis:** Massimiliano Copetti, Alessandra Solari.

**Funding acquisition:** Ambra Mara Giovannetti, Alessandra Solari.

**Investigation:** Ambra Mara Giovannetti.

**Methodology:** Ambra Mara Giovannetti, Massimiliano Copetti, Alessandra Solari.

**Project administration:** Ambra Mara Giovannetti.

**Resources:** Ambra Mara Giovannetti, Alessandra Solari.

**Supervision:** Ambra Mara Giovannetti, Alessandra Solari.

**Visualization:** Ambra Mara Giovannetti.

**Writing – original draft:** Ambra Mara Giovannetti, Kenneth Ian Pakenham, Massimiliano Copetti, Alessandra Solari.

**Writing – review & editing:** Ambra Mara Giovannetti, Kenneth Ian Pakenham, Giovambattista Presti, Maria Esmeralda Quartuccio, Paolo Confalonieri, Roberto Bergamaschi, Monica Grobberio, Massimiliano Di Filippo, Mary Micheli, Giampaolo Brichetto, Francesco Patti, Paola Kruger, Alessandra Solari.

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
