## [Decision Letter · Decision Letter 0]

27 Oct 2021

PONE-D-21-11618

A group resilience training program for people with multiple sclerosis: study protocol of a multi-centre cluster-randomized controlled trial (Multi-READY for MS)

PLOS ONE

Dear Dr. Giovannetti,

Thank you for submitting your manuscript to PLOS ONE. After careful consideration, we feel that it has merit but does not fully meet PLOS ONE’s publication criteria as it currently stands. Therefore, we invite you to submit a revised version of the manuscript that addresses the points raised during the review process.

We look forward to receiving your revised manuscript.

Kind regards,

James Mockridge, PhD

Division Editor

PLOS ONE

Journal Requirements:

Reviewers' comments:

Reviewer's Responses to Questions

**Comments to the Author**

1. Does the manuscript provide a valid rationale for the proposed study, with clearly identified and justified research questions?

Reviewer #1: Yes

Reviewer #2: Yes

2. Is the protocol technically sound and planned in a manner that will lead to a meaningful outcome and allow testing the stated hypotheses?

Reviewer #1: Yes

Reviewer #2: Yes

3. Is the methodology feasible and described in sufficient detail to allow the work to be replicable?

Reviewer #1: Yes

Reviewer #2: Yes

4. Have the authors described where all data underlying the findings will be made available when the study is complete?

Reviewer #1: Yes

Reviewer #2: Yes

5. Is the manuscript presented in an intelligible fashion and written in standard English?

Reviewer #1: Yes

Reviewer #2: Yes

6. Review Comments to the Author

You may also provide optional suggestions and comments to authors that they might find helpful in planning their study.

Reviewer #1: This is an excellently designed protocol and the authors have done their due diligence planning for COVID-19 and how to minimize disruptions in their study. My only suggestion is that the authors consider adding a brief justification as to why a cut-off score of 83 was selected for the CD-RISC 25 in the inclusion/exclusion criteria (line 163).

Reviewer #2: I will focus on methods and reporting. This is a well-written and clear protocol. There are some technical issues I would like to highlight.

Major

1) It wasn't clear to me how the minimisation sampling would work. as far as I know it's either or, can't be both minimisation and randomisation, unless it's some sort of cluster randomisation with the clusters defined by the two variables of interest (centre and dichotomised CDRISK score). Anyway, some clarification is needed here. Also I'd advise you to ensure a process that allows for relatively even balance of cases and controls within each centre, otherwise analyses may become complicated. This may well be the case here, but it was not clear to me as explained.

Minor

1) Ensure analysts are blinded to which group is which (update: I see you have added that to the limitations section, move it to the analysis section or earlier, it is not a limitation)

2) When is it appropriate to use multiple imputation in this context?

3) Will the analyses be controlled for any covariates or is perfect balance across all relevant covariates expected?

4) Need to make it clearer in the outcomes section and the analysis section that the time point of interest is 3-months post-intervention. the immediate post-intervention will be useful in a multiple imputation approach, but would add very little in a complete case analysis.

PS: I reviewed for this funder this summer, but not this work. well done getting a protocol out already.

7. PLOS authors have the option to publish the peer review history of their article (what does this mean?). If published, this will include your full peer review and any attached files.

Reviewer #1: No

Reviewer #2: No

---

## [Author Response · Author response to Decision Letter 0]

15 Dec 2021

Journal Requirements:

RESPONSE 1: We have checked the PLOS ONE's style requirements and modified the files accordingly.

RESPONSE 2: Thanks. We have reported the correct reference in the Reference section: “Qiu J, Shen B, Zhao M, Wang Z, Xie B, Xu Y. A nationwide survey of psychological distress among Chinese people in the COVID-19 epidemic: implications and policy recommendations. Gen Psychiatr. 2020;33(2): e100213.”

We have added the papers published that were previously indicated as “under review” in the reference section.

RESPONSE 3: We checked and corrected the information provided in ‘Funding Information’ and ‘Financial Disclosure’ sections in order for them to match.

RESPONSE 4: Dear editor, this is a protocol paper and no data have been generated yet. Data will be generated and stored during the RCT. The dataset generated and analysed during the current study will be available in the Zenodo repository.

RESPONSE 5: as per your request we moved the information on the Ethical clearance to the Methods section, as reported below: “The protocol received ethical clearance from the ethics committee of the Fondazione IRCCS Istituto Neurologico Carlo Besta (15th April 2020, internal ref: 71; amendment approved 18th November 2020, internal ref: 78) and it has been evaluated by each participating centre’s ethics committee. After obtaining Ethics Committee approvals from all the participating centres, the study will be run at the MS/rehabilitation units of eight university hospitals, research hospitals, general hospitals or community health services across Italy.”

We also modified the abstract accordingly.

Review Comments to the Author

Reviewer #1: This is an excellently designed protocol and the authors have done their due diligence planning for COVID-19 and how to minimize disruptions in their study. My only suggestion is that the authors consider adding a brief justification as to why a cut-off score of 83 was selected for the CD-RISC 25 in the inclusion/exclusion criteria (line 163).

RESPONSE R1.1 = Thank you for the opportunity to clarify this aspect. We discussed the CD-RISC 25 cut off with Prof. Davidson (one of the questionnaire developer) during the design of the pilot study (Giovannetti, A. M., Quintas, R., Tramacere, I., Giordano, A., Confalonieri, P., Messmer Uccelli, M., Solari, A., & Pakenham, K. I. (2020). A resilience group training program for people with multiple sclerosis: Results of a pilot single-blind randomized controlled trial and nested qualitative study. PloS one, 15(4), e0231380). Despite no CD-RISC 25 formal cut-off has been published, he (based on his experience on the use of the scale) suggested us to set this cut-off in order to be sure to include people that had the potential to improve in resilience (avoiding ceiling effect).

To clarify this aspect the following statement was added in the text: “Participant inclusion criteria are: MS diagnosis [56]; age ≥18 years; written informed consent; CD-RISC 25 score < 83, which indicates that the person could still improve his/her level of resilience;”

Reviewer #2: I will focus on methods and reporting. This is a well-written and clear protocol. There are some technical issues I would like to highlight. 

Major

1) It wasn't clear to me how the minimisation sampling would work. as far as I know it's either or, can't be both minimisation and randomisation, unless it's some sort of cluster randomisation with the clusters defined by the two variables of interest (centre and dichotomised CDRISK score). Anyway, some clarification is needed here. Also I'd advise you to ensure a process that allows for relatively even balance of cases and controls within each centre, otherwise analyses may become complicated. This may well be the case here, but it was not clear to me as explained. 

RESPONSE R2.1= Thank you for the opportunity to clarify this important issue. Minimization will be used. We used also the term “randomization”, letting this Reviewer be a little bit confused, since, in general, this technique randomly assigns the first participants, then accounts for the covariates of participants previously enrolled and assigns each new participant to the group that provides better balance. Pure minimization is indeed completely deterministic, that is, we can predict which group the next subject will be enrolled in, provided the factor levels of the new subject are known. This may invalidate the principle of trial blindness and introduce some bias into the trial. To overcome this shortcoming some elements of randomness are incorporated into the minimization algorithm, to make the prediction unlikely. Following this Reviewer suggestion, we revised this section clarifying that patients will be assigned to the treatment arm using the method of minimization. 

“Each centre will collect information on the number of people approached, screened, and eligible prior to random assignment, including reasons for non-enrolment. Participants will then be assigned to the READY or relaxation in a 1:1 ratio using the method of minimization (two factors: Centre and CDRISC score < 50 and ≥ 50) [57]. Treatment assignment will be provided by an independent randomization unit, using a computer-based algorithm [58].”

Minor

1) Ensure analysts are blinded to which group is which (update: I see you have added that to the limitations section, move it to the analysis section or earlier, it is not a limitation)

RESPONSE R2.2 = We added a statement in the analysis section: “Analyses will be carried out in accordance with the pre-specified statistical analysis plan and performed by study personnel blind to the participants’ assignment to the two intervention conditions and to those providing the interventions.”

2) When is it appropriate to use multiple imputation in this context? 

RESPONSE R2.3= We modified the text as follows, in order to specify this aspect: “All group comparisons will be carried out according to the intention-to-treat principle. That is, participants will be analysed in the arm (READY or relaxation) to which they were assigned. We will use multiple imputations for missing data where appropriate (i.e. drop-out, missing item/questionnaire) [85]. In addition, we will carry out a per-protocol analysis and assess the sensitivity of the results to excluding patients who missed three or more READY sessions.”

3) Will the analyses be controlled for any covariates or is perfect balance across all relevant

RESPONSE R2.3= A detailed statistical analysis plan will be developed by the end of the clinical study and the Steering Committee will consider if include or not few covariates (e.g. disease severity-EDSS). Interestingly, the results of previous a study (Giovannetti et al. 2021) on the effectiveness of the READY program in a sample of PwMS showed that no demographic or illness variables predicted the improvements observed at post-intervention or 3-month follow-up, thus suggesting that these variables may play a trivial role in predicting the intervention effect.

Giovannetti, A. M., Solari, A., & Pakenham, K. I. (2021). Effectiveness of a group resilience intervention for people with multiple sclerosis delivered via frontline services. Disability and rehabilitation, 1–11. Advance online publication. https://doi.org/10.1080/09638288.2021.1960441).

We revised the Statistics Section to provide more information about this issue: “Analyses will be carried out in accordance with the pre-specified statistical analysis plan and performed by study personnel blind to the participants’ assignment and to those providing the interventions.

Baseline variables’ standardized mean differences (for clustered data) between arms will be computed to measure potential unbalance. 

Longitudinal changes will be analyzed using repeated measures hierarchical (patients nested in clusters) generalized linear mixed models, accounting for the cluster effect (using random intercepts for clusters) and adjusted for those baseline covariates resulting unbalanced between arms. As reported above, the between-arm difference from baseline (T0) to three-month follow-up (T2) changes in resilience scores is the primary endpoint. To mitigate the risk of inflating Type I error given the small number of clusters, we will use the Kenward-Roger degrees of freedom correction, which does not rely on the assumption of fixed cluster sizes [81].”

4) Need to make it clearer in the outcomes section and the analysis section that the time point of interest is 3-months post-intervention. the immediate post-intervention will be useful in a multiple imputation approach, but would add very little in a complete case analysis.

RESPONSE R2.5= In the Methods section you can find the following statement: “The primary end-point is the between-arm difference from baseline (T0) to three-month follow-up (T2) changes in resilience scores.”

We clarified that the 3-month follow up is our primary endpoint by specifying that even in the following section of the manuscript:

- The Box 1. Panel recommendations: “A six-month post-intervention follow-up should be scheduled in addition to the 3-month follow-up (primary endpoint).

- The Analysis section: “Longitudinal changes will be analyzed using repeated measures hierarchical (patients nested in clusters) generalized linear mixed models, accounting for the cluster effect (using random intercepts for clusters). As reported above, the between-arm difference from baseline (T0) to three-month follow-up (T2) changes in resilience scores is the primary endpoint. To mitigate the risk of inflating Type I error given the small number of clusters, we will use the Kenward-Roger degrees of freedom correction, which does not rely on the assumption of fixed cluster sizes [84].”

---

## [Decision Letter · Decision Letter 1]

6 Apr 2022

A group resilience training program for people with multiple sclerosis: study protocol of a multi-centre cluster-randomized controlled trial (Multi-READY for MS)

PONE-D-21-11618R1

Dear Dr. Giovannetti,

We’re pleased to inform you that your manuscript has been judged scientifically suitable for publication and will be formally accepted for publication once it meets all outstanding technical requirements.

Kind regards,

Carla Pegoraro

Division Editor

PLOS ONE

Additional Editor Comments (optional):

Please note that Reviewer 1 has observed a minor typo. Please amend this during production of your paper.

Reviewers' comments:

Reviewer's Responses to Questions

**Comments to the Author**

1. Does the manuscript provide a valid rationale for the proposed study, with clearly identified and justified research questions?

Reviewer #1: Yes

Reviewer #2: Yes

2. Is the protocol technically sound and planned in a manner that will lead to a meaningful outcome and allow testing the stated hypotheses?

Reviewer #1: Yes

Reviewer #2: Yes

3. Is the methodology feasible and described in sufficient detail to allow the work to be replicable?

Reviewer #1: Yes

Reviewer #2: Yes

4. Have the authors described where all data underlying the findings will be made available when the study is complete?

Reviewer #1: Yes

Reviewer #2: Yes

5. Is the manuscript presented in an intelligible fashion and written in standard English?

Reviewer #1: Yes

Reviewer #2: Yes

6. Review Comments to the Author

You may also provide optional suggestions and comments to authors that they might find helpful in planning their study.

Reviewer #1: The authors have adequately addressed my previous concern. One small note: on page 3, line 54 of the manuscript, it should read "this study" instead of "his study"

Reviewer #2: I am happy with the authors' responses. The paper was reading very well to begin with and only a few clarifications were needed.

7. PLOS authors have the option to publish the peer review history of their article (what does this mean?). If published, this will include your full peer review and any attached files.

Reviewer #1: No

Reviewer #2: No

---

## [Editor Report · Acceptance letter]

22 Apr 2022

PONE-D-21-11618R1 

A group resilience training program for people with multiple sclerosis: study protocol of a multi-centre cluster-randomized controlled trial (Multi-READY for MS) 

Dear Dr. Giovannetti:

I'm pleased to inform you that your manuscript has been deemed suitable for publication in PLOS ONE. Congratulations! Your manuscript is now with our production department. 

Kind regards, 

on behalf of

Dr Carla Pegoraro 

Staff Editor

PLOS ONE